# Implementation considerations for deep learning with diffusion MRI streamline tractography

**Leon Y. Cai**[1]                                        LEON.Y.CAI@VANDERBILT.EDU
[1] *Department of Biomedical Engineering, Vanderbilt University, Nashville, TN, USA*
**Ho Hin Lee**[2]                                        HO.HIN.LEE@VANDERBILT.EDU
[2] *Department of Computer Science, Vanderbilt University, Nashville, TN, USA*
**Nancy R. Newlin**[2]                                  NANCY.R.NEWLIN@VANDERBILT.EDU
**Michael E. Kim**[2]                                      MICHAEL.KIM@VANDERBILT.EDU
**Daniel Moyer**[2]                                        DANIEL.MOYER@VANDERBILT.EDU
**François Rheault**[3]                              FRANCOIS.M.RHEAULT@USHERBROOKE.CA
[3] *Department of Computer Science, Université de Sherbrooke, Sherbrooke, Quebec, Canada*
**Kurt G. Schilling**[4,5]                               KURT.G.SCHILLING.1@VUMC.ORG
[4] *Department of Radiology and Radiological Sciences, Vanderbilt University Medical Center, Nashville, TN, USA*
[5] *Vanderbilt University Institute of Imaging Science, Vanderbilt University, Nashville, TN, USA*
**Bennett A. Landman**[1,2,4,5,6]                      BENNETT.LANDMAN@VANDERBILT.EDU
[6] *Department of Electrical and Computer Engineering, Vanderbilt University, Nashville, TN, USA*

**Editors:** Accepted for publication at MIDL 2023

## Abstract

One area of medical imaging that has recently experienced innovative deep learning advances is diffusion MRI (dMRI) streamline tractography with recurrent neural networks (RNNs). Unlike traditional imaging studies which utilize voxel-based learning, these studies model dMRI features at points in continuous space off the voxel grid in order to propagate streamlines, or virtual estimates of axons. However, implementing such models is non-trivial, and an open-source implementation is not yet widely available. Here, we describe a series of considerations for implementing tractography with RNNs and demonstrate they allow one to approximate a deterministic streamline propagator with comparable performance to existing algorithms. We release this trained model and the associated implementations leveraging popular deep learning libraries. We hope the availability of these resources will lower the barrier of entry into this field, spurring further innovation.

**Keywords:** Diffusion MRI (dMRI) streamline tractography, deep learning, recurrent neural networks, PyTorch

## 1. Introduction

Deep learning has transformed diffusion MRI (dMRI) processing, with many recent studies focusing on streamline tractography with recurrent neural networks (RNNs) (Poulin et al., 2019). Instead of stepping through temporal features to propagate a signal in *time*, these studies step through voxel-based dMRI features to propagate a streamline, or a sequence of points approximating a white matter (WM) tract in the brain, in *space*. However, implementing RNNs to predict sequences of spatial points of arbitrary lengths that may not lie on the voxel-grid with batch-wise backpropagation is non-trivial. Further, an open-source implementation using commonly supported deep learning libraries is not yet widely available. To fill this gap, we detail considerations needed for implementing such a model, assess how one trained with these implementations performs against traditional tractography algorithms, and release the model and associated code implemented in PyTorch (v1.12).

## 2. Methods

**Defining and computing ground truth labels and losses.** We define a batch of $K$ streamlines, $S = s^1, ..., s^K$, as a list of streamlines of non-uniform length. Specifically, we define streamline $s^k$ of length $n^k$ as a list of points, $s^k = \mathbf{x}_1^k, ..., \mathbf{x}_{n^k}^k$, where $\mathbf{x}_i^k$ is a point in continuous 3-dimensional voxel space. We define labels for $\mathbf{x}_i^k$ as the Cartesian unit vector $\Delta \mathbf{x}_i^k = \frac{\mathbf{x}_{i+1}^k - \mathbf{x}_i^k}{||\mathbf{x}_{i+1}^k - \mathbf{x}_i^k||}$. We remove the last point from each streamline so that inputs and labels have the same length, setting $n^k = n^k - 1$. However, as unit vectors have two degrees of freedom, we do not have the RNN directly predict the labels in Cartesian space. Rather, we predict the labels in spherical coordinates as $\Delta \hat{\mathbf{x}}_i^k = (\phi_i^k, \theta_i^k)$ and convert to Cartesian as $\Delta \hat{\mathbf{x}}_i^k = (\sin \phi_i^k \cos \theta_i^k, \sin \phi_i^k \sin \theta_i^k, \cos \phi_i^k)$ prior to loss computation. We utilize a cosine similarity loss for each point of $s^k$, $\mathcal{L}(\Delta \hat{\mathbf{x}}_i^k, \Delta \mathbf{x}_i^k) = 1 - \frac{\langle \Delta \hat{\mathbf{x}}_i^k, \Delta \mathbf{x}_i^k \rangle}{||\Delta \hat{\mathbf{x}}_i^k|| ||\Delta \mathbf{x}_i^k||}$. Streamlines can be propagated from the $i$th point to the next as $\hat{\mathbf{x}}_{i+1}^k = \mathbf{x}_i^k + \gamma \Delta \hat{\mathbf{x}}_i^k$ where $\gamma$ is the step size.

**Differentiably sampling dMRI features off the voxel grid.** $\mathbf{x}_i^k$, defined as a 3-dimensional coordinate in voxel space, provides little utility for efficiently querying dMRI information at its location off the voxel grid. Thus, we instead convert each $\mathbf{x}_i^k$ to $\mathbf{c}_i^k$, an 11-dimensional vector. Considering $\mathbf{x}_i^k$ as an off-grid point contained within a lattice of 8 on-grid points, the first 3 elements of $\mathbf{c}_i^k$ are the distance of $\mathbf{x}_i^k$ from the lowest lattice point along all 3 spatial axes in voxel space, $\mathbf{x}_i^k - \lfloor \mathbf{x}_i^k \rfloor$. The remaining 8 elements are the linear indices of the 8 on-grid points in the image volume. With these 11 values, the lattice values can be queried and interpolated trilinearly to obtain off-grid features for each point in $s^k$ as $\mathbf{q}_i^k = dMRI(\mathbf{c}_i^k)$ (Kang, 2006). As trilinear interpolation is differentiable, this allows for end-to-end training between input voxel grids and output losses at points off the grid.

**Organizing data during training.** As an example, we assume each $\mathbf{q}_i^k$ is a 45-dimensional feature vector, as is commonly the case if the dMRI grid is a grid of fiber orientation distribution (FOD) spherical harmonic (SH) coefficients. Thus, $S$ can be represented as a list of length $K$ where each $s^k$ is a matrix of size $n^k \times 45$. However, the variability of $n^k$ across $S$ is inefficient for the tensor-based parallelization frameworks utilized by deep learning libraries. Thus, we convert $S$ into a "padded packed" tensor for training.

When aligned by the first element of each $s^k$, $S$ can be "padded" with zeros to a tensor of size $M \times K \times 45$, where $M = \max(n^1, ..., n^K)$ is the length of the longest streamline in the batch. This padded tensor can then be "packed" to a tensor of size $N \times 45$, where $N = \sum_{k=1}^{K} n^k$. The packed formulation allows for batch-wise steps in recurrent neural networks for input sequences of different lengths, and the padded formulation allows for easier querying of specific points in their corresponding streamlines for loss aggregation. Both these operations and their inverses are natively supported in PyTorch.

The network predictions are also packed tensors of size $N \times 3$ after conversion from spherical to Cartesian coordinates. To compute the batch-wise loss, we convert the packed predictions to padded representations of size $M \times K \times 3$, use a mask to ignore the padding, and average the loss across all the streamline points as $\frac{1}{N} \sum_{k=1}^{K} \sum_{i=1}^{n^k} \mathcal{L}(\Delta \hat{\mathbf{x}}_i^k, \Delta \mathbf{x}_i^k)$. For efficiency, we compute masks and save the labels in padded form before training.

**Parallelizing inference.** Unlike traditional tractography algorithms which parallelize tracking on the streamline level, RNNs must parallelize on the point level. In other words, each step of the RNN must advance all streamlines in a batch, as outlined in algorithm 1.

---

**Algorithm 1:** Parallelizing inference with a padded tensor where $M = 1$

---

1. $\mathbf{x}_i^1, ..., \mathbf{x}_i^K$ (size $1 \times K \times 3$) are the heads of $K$ actively propagating streamlines in a padded tensor. These points are seeded arbitrarily when $i = 1$.
2. Convert $\mathbf{x}_i^1, ..., \mathbf{x}_i^K$ to $\mathbf{c}_i^1, ..., \mathbf{c}_i^K$ (size $1 \times K \times 11$).
3. Sample $\mathbf{q}_i^1, ..., \mathbf{q}_i^K$ (size $1 \times K \times 45$) off-grid from $\mathbf{c}_i^1, ..., \mathbf{c}_i^K$.
4. Compute $\Delta\hat{\mathbf{x}}_i^1, ..., \Delta\hat{\mathbf{x}}_i^K$ (size $1 \times K \times 3$) with the RNN from $\mathbf{q}_i^1, ..., \mathbf{q}_i^K$.
5. Compute $\hat{\mathbf{x}}_{i+1}^1, ..., \hat{\mathbf{x}}_{i+1}^K = \mathbf{x}_i^1 + \gamma\Delta\hat{\mathbf{x}}_i^1, ..., \mathbf{x}_i^K + \gamma\Delta\hat{\mathbf{x}}_i^K$ (size $1 \times K \times 3$).
6. Set $\mathbf{x}_i^1, ..., \mathbf{x}_i^K = \hat{\mathbf{x}}_{i+1}^1, ..., \hat{\mathbf{x}}_{i+1}^K$ and repeat.

---

This approach allows arbitrary stopping criteria to be evaluated for each streamline head independently, after which it can be taken off the tensor, speeding up propagation for the remaining streamlines. Since batches have a set size $K$, once all streamlines meet criteria, new batches can be initialized and propagated until the desired number of streamlines are generated. Last, $K$ can vary, making this approach adaptable to different GPU capacities.

## 3. Results and Discussion

With these considerations, we train an RNN streamline propagator on dMRI data from the Human Connectome Project to approximate the deterministic SDStream tractography algorithm (Tournier et al., 2007) as described by Cai et al. (2023). Briefly, we use a multi-layer perceptron- and gated recurrent unit-based architecture with 4.2 million parameters, taking dMRI FODs represented on the voxel grid with 45 even-order SH coefficients as input. Compared to SDStream, we find similar recovery of WM bundles between our method and the iFOD2 probabilistic propagator (Tournier et al., 2010) (Figure 1).

We release this model and the associated code (github.com/MASILab/STrUDeL) to spur further innovations in this field. We note these implementations are currently limited to deterministic propagators, and probabilistic ones would require reparameterization.

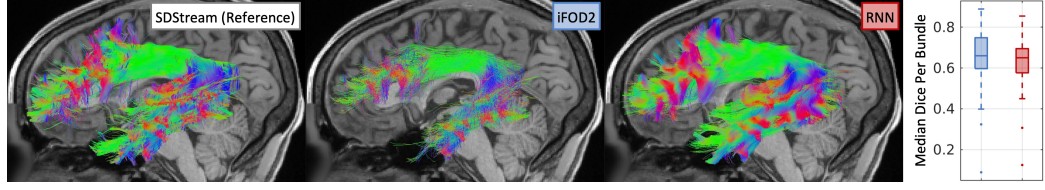

Figure 1: Compared to reference, representative iFOD2 and RNN left arcuate fasciculii are visually similar as are the median Dice coefficients across subjects per bundle.

## Acknowledgments

This work was supported by ACCRE at Vanderbilt; NSF 2040462; and NIH intramural and 5R01EB017230, U34DK123895, P50HD103537, U54HD083211, K01EB032898, and T32GM007347; and does not necessarily represent the official views of the NIH or NSF.

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
