# OpenReview forum: "Implementation considerations for deep learning with diffusion MRI streamline tractography"
_MIDL.io/2023/Short_Paper_Track — MIDL 2023 Short paper track Poster_

### Official Review · Reviewer_68DW · 2023-04-19
**a useful open-source implementation of dMRI streamline using RNNs BUT the link did not work?**

**Rating:** 6
**Confidence:** 3

**Review:**

Diffusion MRI is an imaging modality that measures the anisotropic diffusion of water molecules within 3d voxels. In the brain water molecules tend to diffuse along neural fibers. Streamlines are imaginary curves that can be obtained by following the "stream" of preferred diffusion orientation in the hope of solving the tractography problem of inferring possible neural tracts

The article presents a deep learning approach to diffusion MRI streamline tractography using recurrent neural networks. The authors discuss implementation considerations and offer some evidencet that their model approximates a deterministic streamline technique.

Pros:
* The study demonstrates use of deep learning, specifically RNNs, for dMRI streamline tractography.
* The authors provide some considerations for implementing tractography with RNNs.
* The model shows comparable performance to existing algorithms, making it a viable alternative for dMRI streamline tractography.

Cons:
* THE PROVIDED IMPLEMENTATION LINK DID NOT WORK: github.com/MASILab/STrUDeL
this terribly weakens the main benefit of this short paper and should be fixed before acceptance

Ideas for Related and Future Work:
* probabilistic propagators (as suggested by the authors)
* explore possible different network architectures
* more quantitative validation against comparable existing techniques
* impact on clinical decision making and outcomes

---

### Official Review · Reviewer_2caP · 2023-04-22
**Interesting details discussed, but evaluation and literature are lacking**

**Rating:** 7
**Confidence:** 3

**Review:**

The paper is well-written and clear to follow, and presents interesting aspects of deep learning for tractography.


Pros:
- Interesting discussion of implementation details and motivation thereof that is often brushed over in other papers
- Promising preliminary results

Cons:
- Discussion of other deep learning approaches for tractography is lacking, beyond Poulin 2019 there is also Track-to-learn <https://github.com/scil-vital/TrackToLearn>, DeepTract <https://github.com/itaybenou/DeepTract> and probably others.
- The repo indicated (<https://github.com/MASILab/STrUDeL>) is private and cannot be checked for review.
- Evaluation is lacking - beyond Dice it would be good to have a bundle-level evaluation in terms of sensitivity, specificity (see Tractometer paper in MIA). Also, what is the impact on downstream whole-brain connectomes, how different are they from the oridinal SDStream ones in terms of topological properties etc? Finally, it would be good to evaluate on lower-quality data than HCP - most MR systems don't have gradients that are as good.